# Theoretical Justification of Structural, Magnetoelectronic and Optical Properties in QFeO_3_ (Q = Bi, P, Sb): A First-Principles Study

**DOI:** 10.3390/mi14122251

**Published:** 2023-12-17

**Authors:** Amna Parveen, Zeesham Abbas, Sajjad Hussain, Shoyebmohamad F. Shaikh, Muhammad Aslam, Jongwan Jung

**Affiliations:** 1College of Pharmacy, Gachon University, No. 191, Hambakmeoro, Yeonsu-gu, Incheon 21936, Republic of Korea; 2Department of Nanotechnology and Advanced Materials Engineering, Sejong University, Seoul 05006, Republic of Korea; zeeshamabbas035@gmail.com (Z.A.); shussainawan@gmail.com (S.H.); 3Department of Chemistry, College of Science, King Saud University, P.O. Box 2455, Riyadh 11451, Saudi Arabia; 4Institute of Physics and Technology, Ural Federal University, Mira Str. 19, 620002 Yekaterinburg, Russia

**Keywords:** first-principles calculations, DFT, multiferroics, optical properties, perovskites, magnetic properties

## Abstract

One of the primary objectives of scientific research is to create state-of-the-art multiferroic (MF) materials that exhibit interconnected properties, such as piezoelectricity, magnetoelectricity, and magnetostriction, and remain functional under normal ambient temperature conditions. In this study, we employed first-principles calculations to investigate how changing pnictogen elements affect the structural, electronic, magnetic, and optical characteristics of QFeO_3_ (Q = Bi, P, SB). Electronic band structures reveal that BiFeO_3_ is a semiconductor compound; however, PFeO_3_ and SbFeO_3_ are metallic. The studied compounds are promising for spintronics, as they exhibit excellent magnetic properties. The calculated magnetic moments decreased as we replaced Bi with SB and P in BiFeO_3_. A red shift in the values of $\varepsilon_{2}(\omega )$ was evident from the presented spectra as we substituted Bi with Sb and P in BiFeO_3_. QFeO_3_ (Q = Bi, P, SB) showed the maximum absorption of incident photons in the visible region. The results obtained from calculating the optical parameters suggest that these materials have a strong potential to be used in photovoltaic applications.

## 1. Introduction

Multiferroic (MF) materials have attracted the interest of researchers due to their technological applications as ferroelastic, ferroelectric, and (anti)ferromagnetic materials in a single crystalline phase that gives these materials added advantages over others. Specifically, magnetoelectric (ME) coupling is the best example of multiferroism [1,2,3]. Therefore, magnetic ordering can be produced in these materials by an electric field, and electric polarization can be produced by a magnetic field [4,5]. The existence of ME coupling is the primary condition for the existence of multiferroism; however, a material cannot be categorized as an MF material even if both properties are present in the material. These materials can act as potential candidates for technological applications in biomedicines, data storage devices, sensors, magnetic readers, actuators, spintronics, information processing and storage, tunneling, and others [6,7,8,9,10,11,12,13,14,15,16]. Unfortunately, magnetic and ferroelectric behavior exists in few compounds above room temperature. BiFeO_3_ (BFO) is the most capable multiferroic material as it shows both properties simultaneously. In recent years, numerous research articles have been published on multiferroic BiFeO_3_ (bismuth ferrite) due to its extraordinary nature of having ferroelectric and (anti)ferromagnetic properties at room temperature [9,17,18,19]. Researchers have focused on single-phase BiFeO_3_ because of its high antiferromagnetic Néel temperature (up to 640 K) and ferroelectric transition temperature (1100 K) [9,17,18,19]. However, feeble magnetic properties [20] and high leakage current value [21] are some of the main challenges associated with this material. The creation of secondary phases (Bi_2_Fe_4_O_9_ and Bi_25_FeO_40_) and a narrow window for thermal stability are the main hurdles in the development of unpolluted single-phased BiFeO_3_ [22,23].

Researchers studied ME coupling steadily from it was first described in 1961 until 2003, and then the high ferroelectricity of BFO films and energetic coupling among magnetic and ferroelectric properties in TbMn_2_O_5_ and TbMnO_3_ was discovered [23,24]. Thereafter, studies focusing on this class of materials (MF materials) increased drastically. Undoubtedly, because of the excellent ferroelectric properties, prominent ME coupling, and the facilities to synthesize it, it can easily be grown on SrTiO_3_ substrates using epitaxial growth [25]. BiMnO_3_, TbMnO_3_, NiTiO_3_, PbTiO_3_, FeTiO_3_, YMnO_3_, PbVO_3_, PbNiO_3_, BaNiF_4_, TbMn_2_O_5_, LuFe_2_O_4_, and Ca_3_CoMnO_6_ are some other examples of multiferroic compounds [26,27,28,29,30,31,32]. Numerous MF compounds are known nowadays, but the crystal structure of these materials is the main hurdle in their development because this property is limited to perovskite, corundum-ordered structures (LiNbO_3_ structure—R3c), and ilmenite (R3) [33,34,35]. Ferroelectric properties in these compounds arise due to strongly compressed layers of O atoms, the degree of distortion at octahedral sites, exchange along the c axis among A and B cations [36], and the intrinsic occurrence of ordering vacancies [37,38,39,40].

Recent work on MF compounds and particularly ME coupling explains ways to improve these features through the development of double perovskites [41,42], heterojunction [43,44], doping [45,46], co-doping [47,48], and modulation of different properties [49,50]. In these cases, theoretical methods have been employed to address the magnetic, optical, electronic, and structural properties of the investigated compounds; however, experimental studies have explained the improvement in the magnetoelectric coupling of the investigated compounds. In this manuscript, density functional theory (DFT) calculations were used to study the structural, electronic, optical, and magnetic properties of QFeO_3_ (X = Bi, P, Sb).

## 2. Materials and Methods

The main motivation to perform this study was to use the FP-LAPW+lo (full-potential linearized augmented plane wave plus local orbital) method [51,52] to obtain the solution to the Kohn–Sham equations with generalized gradient approximation (GGA) as executed in WIEN2K (version 14.0) code [53,54] in the context of DFT for perovskite materials QFeO_3_ (Q = Bi, P, Sb). In these calculations, the exchange correlation potential was treated by the GGA approximation proposed by Perdew Burke and Emzerhof [55]. It is a well-known fact that the optical and electronic properties of systems with strongly correlated electrons, especially those with incomplete 3d, 4f, and 5f orbitals, cannot be calculated accurately by simple GGA or local density (LDA) approximation [56]. Researchers have proposed numerous theoretical methods to overcome this problem. In this manuscript, the problem of strongly correlated potentials was overcome by adding the Hubbard Hamiltonian to GGA functions (GGA+U). In these calculations, a set of U values (3.0 eV, 5.0 eV, 7.0 eV, and 9.0 eV) was employed to obtain optimized results. We chose to present results calculated with U = 5.0 eV from the mentioned list of energies. The calculated TDOS spectra for QFeO_3_ (Q = Bi, P, Sb) at all U values are presented in the manuscript. The rest of the results, including magnetic properties, energy band structures, and optical properties, are presented only for U = 5.0 eV.

Unit cells were divided into two parts when working with the FP-LAPW method, i.e., the interstitial region (IR) and atomic spheres (muffin tins). The IR is a region other than the muffin tin spheres. Two completely diverse basis sets expand the wave function in these regions. The wave functions in the muffin tins and IRs are expanded with the help of the atomic-like function, as shown in Equation (1), and the plane wave basis, as shown in Equation (2).
(1)$$V\left(r\right)=\sum_{l,m}V_{lm}(r)Y_{lm}(r)$$
(2)$$V\left(r\right)=\sum_{k}V_{k}e^{i\cdot \vec{r}}$$

Here, $\vec{k\;}$ and $V_{k}$ are the Fermi wave vectors in the irreducible Brillouin zone (IBZ) and the scalar amplitude of the plane wave, respectively. The relativistic and semi-relativistic methods are used for core and valance electrons, respectively. Choosing the best input factors is essential in first-principles calculations. Numerous input factors were set in such a way that they yielded the most precise results and ensured excellent convergence of total energy. Suitable values of $R_{MT}$ were used to ensure the convergence of total energy and to prevent the leakage of charge from the core states of these compounds. The following $R_{MT}$ values were selected: 1.63, 1.89, 2.5, 2.37, and 2.5 for O, Fe, Bi, P, and Sb, respectively. The values of $l_{max}$, $G_{max}$, and $R_{MT}\times k_{max}$ were 10, 12, and 7.0 respectively. Here, $l_{max}$, $R_{MT}$, and $k_{max}$ were used for the maximum value of angular momentum, the smallest muffin tin radius, and the maximum value of the Fermi wave vector, respectively. Self-consistent field calculations for the optical and electronic properties of QFeO_3_ (Q = Bi, P, Sb) were performed using a k-mesh of 500 points with a modified tetrahedron approach. The rhombohedral crystalline structures of QFeO_3_ (Q = Bi, P, Sb) with the space group R3c are presented in Figure 1.

## 3. Results and Discussion

### 3.1. Electronic Properties

#### 3.1.1. Electronic Band Structures

Studying energy band structure is very useful, as numerous physical properties of crystalline solids are associated with energy band structure. Therefore, knowledge of energy band gaps is crucial because of dexterous applications in electromagnetic, optoelectronic, and magneto-optic devices. Materials with collinear conduction band minima (CBM) and valance band maxima (VBM) are known as direct band gap semiconductors, and others are known as indirect band gap semiconductors. The response of indirect band gap materials to optical excitations is very weak compared with that of direct band gap materials. In addition to photons and electrons, phonons also participate in optical transitions to preserve the momentum in indirect band gap materials, and this is the reason that researchers prefer direct band gap materials instead of indirect band gap materials for applications in optoelectronic devices.

Energy band calculations were performed for QFeO_3_ (Q = Bi, P, Sb) along the highly symmetric axis (R, Γ, X, M, Γ) in the irreducible Brillouin zone (IBZ) for both majority and minority spins. Calculated energy band structures with U = 5.0 eV are presented in Figure 2 for QFeO_3_ (Q = Bi, P, Sb). In Figure 2, the dotted line shows the Fermi level $E_{F}$, and the region below the dotted line is known as the valance band, whereas the region above the dotted line is known as the conduction band. The calculation reveals that QFeO_3_ compounds (Q = Bi, P, Sb) are indirect band gap semiconductors. The calculated values of the energy bandgaps for QFeO_3_ (Q = Bi, P, Sb) for various values of U are tabulated in Table 1. The different behaviors of the calculated energy band structures for the majority and minority spins showed spin anisotropy, which was responsible for the magnetic moments in QFeO_3_ (Q = Bi, P, Sb).

#### 3.1.2. Density of States

The density of states (total (TDOS) and partial (PDOS)) calculated by spin-polarized (SP) calculations for QFeO_3_ (Q = Bi, P, Sb) are discussed in this section of the manuscript. The calculated TDOS spectra for QFeO_3_ (Q = Bi, P, Sb) at all U values (3.0 eV, 5.0 eV, 7.0 eV, and 9.0 eV) are presented in Figure 3. A remarkable difference was present in the spectra of spin-polarized TDOS QFeO_3_ (Q = Bi, P, Sb) for both majority and minority spins presented in Figure 3 from −6.0 to 6.0 eV. Orbitals below −6.0 eV are known as core states, and they do not have a significant impact on the physical properties of compounds, which means that core electrons are not used to determine magnetic, optical, thermal, and electronic properties. Likewise, orbitals present between −6.0 to −4.0 are known as semi-core electrons, and they do not have a significant impact on the physical properties of compounds either. One can note that the DOS values of the entire energy range are different for majority and minority spins, which shows that these materials possess non-zero total magnetic moments. The magnetic moment for QFeO_3_ (Q = Bi, P, Sb) above the Fermi level (conduction band) is present due to the disproportion of spin in the minority spin, as the DOS in the minority spin is dominant compared with the DOS in the majority spin.

The spectra of the PDOS presented in Figure 4 show that the major contribution to the valance band of BiFeO_3_ came from O and Fe atoms, and some minor contributions from Bi atoms were also present. One can notice from the spectra of PDOS that the O-${2p}^{4}$, Bi-${6s}^{2}$, and Bi-$6p^{3}$ orbitals had major contributions to the valance band, whereas the Fe-${3d}^{6}$ orbitals had some minor contributions. In the conduction band, major contributions were present due to the Fe atoms in the minority spin channel, and some minor contributions from the Bi and O-atoms were also present. One can notice from the spectra of the PDOS that Fe-${3d}^{6}$ orbitals had major contributions to the conduction band, whereas the Bi-$6p^{3}$ and O-${2p}^{4}$ orbitals had some minor contributions.

The spectra of the PDOS presented in Figure 5 show that the major contribution to the valance band of PFeO_3_ came from O- and P-atoms, and some minor contributions from Fe atoms were also present. One can notice from the spectra of the PDOS that the O-${2p}^{4}$ and P-$3p^{3}$ orbitals had major contributions to the valance band, whereas Fe-${3d}^{6}$ orbitals also had some minor contributions i. Major contributions in the conduction band were present due to P and Fe atoms, whereas minor contributions from O atoms were also present. One can notice from the spectra of the PDOS that in the conduction band, the P-$3p^{3}$ and Fe-${3d}^{6}$ orbitals had major contributions and minor contributions from O-${2p}^{4}$.

The spectra of the PDOS presented in Figure 6 show that the major contribution in the valance band of SbFeO_3_ came from O atoms, and some minor contributions from Sb and Fe atoms were also present. One can notice from the spectra of the PDOS that the O-${2p}^{4}$ orbitals had major contributions, whereas the Sb-${5p}^{3}$ and Fe-${3d}^{6}$ orbitals also had minor contributions in the valance band. In the conduction band, major contributions were present from Fe and Sb atoms in the minority spin channel, and some minor contributions from O atoms were also present. One can notice from the spectra of the PDOS that Fe-${3d}^{6}$, Sb-${5s}^{2}$, and Sb-${5p}^{3}$ orbitals had major contributions, whereas O-${2p}^{4}$ orbitals had some minor contributions in the conduction band.

### 3.2. Magnetic Properties

Magnetic moments were also calculated in this study with the GGA+U approach. A noticeable difference was present in the spectra of PDOS and TDOS for QFeO_3_ (Q = Bi, P, Sb) near the Fermi level in different spins. This difference was the origin of magnetic behavior in the compounds above. Information regarding the electronic states with a major impact on magnetic properties can be obtained from the spectra of the PDOS. The spectra of the PDOS presented in Figure 4, Figure 5 and Figure 6 show that the magnetic moment in the valance band was mainly present due to the spin imbalance of O atoms from −4.0 to −1.0 eV; however, magnetic moment in the conduction band was mainly present due to spin imbalance of Fe atoms in the negative region between 2.0 and 4.0 eV. The spectra of PDOS for BiFeO_3_ presented in Figure 5 show that magnetic moment in the valance band was mainly present due to spin imbalance of the Bi-${6p}^{3}$, Fe-${3d}^{6}$, and O-${2p}^{4}$ orbitals. However, the magnetic moment in the conduction band was mainly present due to the spin imbalance of the Fe-${3d}^{6}$ and Bi-${6p}^{3}$ orbitals.

The spectra of the PDOS for PFeO_3_ presented in Figure 6 show that magnetic moment in the valance band was mainly present due to the spin imbalance of the P-${3p}^{3}$ and O-${2p}^{4}$ orbitals. However, the magnetic moment in the conduction band was mainly present due to the spin imbalance of the Fe-${3d}^{6}$, P-${3p}^{3}$, and O-${2p}^{4}$ orbitals. The spectra of the PDOS for SbFeO_3_ presented in Figure 6 show that the magnetic moment in the valance band was mainly present due to the spin imbalance of the Sb-${5p}^{3}$ and O-${2p}^{4}$ orbitals. However, the magnetic moment in the conduction band was mainly present due to the spin imbalance of the Fe-${3d}^{6}$, Sb-${5p}^{3}$, and O-${2p}^{4}$ orbitals. The calculated total and partial magnetic moments are presented in Table 2. We can observe from Table 2 that Fe atoms had major contributions in all these compounds.

### 3.3. Optical Properties

The response of the medium in the presence of external electromagnetic radiation can be explained with the help of the complex dielectric function *ε*(*ω*). The following equation can be used to calculate ε(ω):(3)$$\varepsilon \left(\omega \right)=\varepsilon_{1}(\omega )+i\varepsilon_{2}(\omega )$$

Here, $\varepsilon_{1}(\omega )$ and $\varepsilon_{2}(\omega )$ are known as the real and imaginary parts of the complex dielectric function. The imaginary part $\varepsilon_{2}(\omega )$ of the complex dielectric function explains the absorptive behavior of the medium and is directly associated with energy band structure. The following equation can be used to calculate $\varepsilon_{2}(\omega )$:(4)$$\varepsilon_{2}\left(\omega \right)=\frac{8\pi^{2}e^{2}}{\omega^{2}m^{2}}\sum_{n}\sum_{n^{\prime }}\int {\left|P_{nn^{\prime }}^{v}\left(k\right)\right|}^{2}f_{kn}(1-f_{nn^{\prime }})\delta (E_{n}^{k}-E_{n^{\prime }}^{k}-\hbar \omega )\frac{d^{3}k}{{\left(2\pi \right)}^{3}}$$

Here, *m*, *e*, $E_{n}^{k}\left(k\right)$, $f_{kn}$, and $P_{nn^{\prime }}^{v}(k)$ are used to specify the mass of the electron, the charge of the electron, the energy of one electron, the Fermi–Dirac distribution, and the projection of momentum dipole matrix elements, respectively. These matrix elements are described between the initial and final stages of the electric field in the v direction. The calculated spectra of $\varepsilon_{2}(\omega )$ for QFeO_3_ (Q = Bi, P, Sb) are presented in Figure 7 for both the minority and majority spins. The heights and shapes of the peaks in the spectra of $\varepsilon_{2}(\omega )$ for QFeO_3_ (Q = Bi, P, Sb) were different in the minority and majority spin channels. The point form where peaks start to originate is known as the threshold energy of the material. The threshold energies of $\varepsilon_{2}(\omega )$ calculated from Figure 7 for QFeO_3_ (Q = Bi, P, Sb) are presented in Table 3. In both spin channels, PFeO_3_ showed the maximum absorption of photons in the visible and infrared region (IR), and BiFeO_3_ had the lowest absorption in the visible region. One can observe from Figure 7 that the spectra of $\varepsilon_{2}(\omega )$ shifted toward higher energies as P was substituted for Sb and Bi. The origin of different peaks in the spectra of $\varepsilon_{2}(\omega )$ can be identified by the spectra of the PDOS. The first peak in the spectra of $\varepsilon_{2}(\omega )$ occurred due to electronic transitions from the O-${2p}^{4}$ orbitals to Bi-${6p}^{3}$, P-${3p}^{3}$, and Sb-${5p}^{3}$ for BiFeO_3_, PFeO_3_, and SbFeO_3_, respectively.

The following expression can be used to calculate the real part $\varepsilon_{1}(\omega )$ of the complex dielectric function using the $\varepsilon_{2}(\omega )$ calculated by the Kramers–Kronig transformation:(5)$$\varepsilon_{1}\left(\omega \right)=1+\frac{2}{\pi }P\int_{0}^{\infty }\frac{\omega^{\prime }\varepsilon_{2}(\omega^{\prime })}{\omega^{\prime 2}-\omega^{2}}d\omega^{\prime }$$

The real part $\varepsilon_{1}(\omega )$ of the complex dielectric function explains the dispersive behavior of the medium and is inversely associated with energy band structure. The calculated spectra of $\varepsilon_{1}(\omega )$ for QFeO_3_ (Q = Bi, P, Sb) are presented in Figure 7 for both the minority and majority spins. The heights and shapes of the peaks in the spectra of $\varepsilon_{1}(\omega )$ for QFeO_3_ (Q = Bi, P, Sb) were different in the minority and majority spin channels. The most significant quantity in the spectra of $\varepsilon_{1}(\omega )$ is its static value $\varepsilon_{1}(0)$, also known as the zero-frequency limit. The static values of the dielectric constant $\varepsilon_{1}(0)$ calculated from Figure 7 for QFeO_3_ (Q = Bi, P, Sb) are presented in Table 4. The following relationship can be used to relate the energy band gap and the static dielectric constant.
(6)$$\varepsilon_{1}\left(0\right)\approx 1+{\left(\frac{\hbar \omega_{p}}{E_{g}}\right)}^{2}$$

One can observe that values of $\varepsilon_{1}(0)$ increased as Bi was substituted for Sb and P, confirming its inverse relationship with the energy band gap. There was an increasing trend in the spectra of $\varepsilon_{1}(\omega )$ after the zero-frequency limit except for PFeO_3_ in the majority spin channel. The spectra of $\varepsilon_{1}(\omega )$ reached their peak values, then a sharp dip was observed in the spectra, and finally, these curves entered the negative region at certain values. The frequencies at which these curves enter the negative region are known as the plasmon frequencies of the compounds. The energy ranges at which the spectra remained in the negative region showed complete attenuation of the incident beam of photons.

Knowledge of $\varepsilon_{2}(\omega )$ and $\varepsilon_{1}(\omega )$ can be used to calculate the rest of the optical parameters such as the extinction coefficient K(ω), refractive index n(ω), reflectivity coefficient R(ω), absorption coefficient I(ω), energy loss function L(ω), and the real part of optical conductivity σ(ω). The following equation can be used to calculate K(ω) by utilizing the knowledge of $\varepsilon_{2}(\omega )$ and $\varepsilon_{1}(\omega )$:(7)$$K\left(\omega \right)={\left[\frac{\sqrt{\varepsilon_{1}^{2}(\omega )+\varepsilon_{2}^{2}(\omega )}}{2}-\frac{\varepsilon_{1}(\omega )}{2}\right]}^{1/2}$$

The extinction coefficient K(ω) showed features similar to those of $\varepsilon_{2}(\omega )$, which means that it also tells us about the absorptive behavior of the compound. The calculated spectra of K(ω) for QFeO_3_ (Q = Bi, P, Sb) are presented in Figure 8 for both the minority and majority spins. The heights and shapes of the peaks in the spectra of K(ω) for QFeO_3_ (Q = Bi, P, Sb) were different in the minority and majority spin channels. The point form where the peaks originate is known as the threshold energy of the material. The threshold energies of K(ω) calculated from Figure 8 for QFeO_3_ (Q = Bi, P, Sb) are presented in Table 3. In both spin channels, PFeO_3_ showed the maximum absorption of photons in the visible and infrared region (IR), and BiFeO_3_ had the lowest absorption in the visible region. One can observe from Figure 9 that the spectra of K(ω) shifted toward higher energies as P was substituted for Sb and Bi.

The following equation can be used to calculate n(ω) by utilizing the knowledge of $\varepsilon_{2}(\omega )$ and $\varepsilon_{1}(\omega )$:(8)$$n\left(\omega \right)={\left[\frac{\sqrt{\varepsilon_{1}^{2}(\omega )+\varepsilon_{2}^{2}(\omega )}}{2}+\frac{\varepsilon_{1}(\omega )}{2}\right]}^{1/2}$$

The refractive index n(ω) is an important parameter to check the suitability of an optical material for its technological applications in detectors, waveguides, photonic crystals, and solar cells. The refractive index n(ω) shows features similar to those of $\varepsilon_{1}(\omega )$, which means that it also tells us about the dispersive behavior of the compound. The calculated spectra of n(ω) for QFeO_3_ (Q = Bi, P, Sb) are presented in Figure 8 for both the minority and majority spins. The heights and shapes of the peaks in the spectra of n(ω) for QFeO_3_ (Q = Bi, P, Sb) were different in the minority and majority spin channels. The most significant quantity in the spectra of n(ω) is its static value n(0), also known as the zero-frequency limit. The static values of n(0) calculated from Figure 8 for QFeO_3_ (Q = Bi, P, Sb) are presented in Table 4. The following relationship can be used to relate static values of the dielectric constant with the static values of the refractive index:(9)$$n\left(0\right)=\sqrt{\varepsilon_{1}\left(0\right)}$$

One can observe that values of n(0) increased as Bi was substituted with Sb and P. There was an increasing trend in the spectra of n(ω) after the zero-frequency limit except for PFeO_3_ in the majority spin channel. The spectra of n(ω) reached their peak values, then a sharp dip was observed in the spectra, and finally, these curves became less than unity (n=c/v) at certain energies. This shows an unnatural phenomenon in which the speed of light in vacuum (c) is less than the group velocities of the incident photons ($V_{g}$). This means that the medium shows non-linear behavior instead of linear behavior, or we can say that the medium becomes superluminal [57].

The following expression explains the reflectivity coefficient R(ω) in terms of n(ω) and K(ω) [58]:(10)$$R\left(\omega \right)=\frac{{\left(1-n\right)}^{2}+K^{2}}{{\left(1+n\right)}^{2}+K^{2}}$$

The calculated values of the complex dielectric function ε(ω) can also be used to explain the behavior of R(ω) with the help of the following expression [58]:(11)$$R\left(\omega \right)=\left|\frac{1-\sqrt{\varepsilon (\omega )}}{1+\sqrt{\varepsilon (\omega )}}\right|$$

The calculated spectra of R(ω) for QFeO_3_ (Q = Bi, P, Sb) are presented in Figure 9 for both the minority and majority spins. The heights and shapes of the peaks in the spectra of R(ω) for QFeO_3_ (Q = Bi, P, Sb) were different in the minority and majority spin channels. The most significant quantity in the spectra of R(ω) is its static value R(0), also known as the zero-frequency limit. The static values of R(0) calculated from Figure 9 for QFeO_3_ (Q = Bi, P, Sb) are presented in Table 4. The calculated results showed that the reflectivity R(ω) of PFeO_3_ was greater than that of BiFeO_3_ and SbFeO_3_. From spectra of R(ω), it can be concluded that PFeO_3_ can be used in applications in which a maximum of 60% reflection of incident photons is needed. BiFeO_3_ and SbFeO_3_ are poorly reflecting materials in the entire energy range. An interesting behavior of reflectivity can clearly be seen in Figure 9, in which the peak value of reflectivity occurs in the region where $\varepsilon_{1}(\omega )$ becomes negative because the reflectivity of the compound and metallicity are directly associated with each other. The region where $\varepsilon_{1}(\omega )$ becomes negative reveals the metallic nature of compounds [59]. The reflectivity for BiFeO_3_, PFeO_3_, and SbFeO_3_ started from 12.7%, 28.8%, and 16.6%, respectively, for the majority spin channel. However, reflectivity in the minority spin channels for BiFeO_3_, PFeO_3_, and SbFeO_3_ started from 15.6%, 24.8%, and 18.7%, respectively. The reflectivity attained its maximum values of approximately 42%, 58%, and 45% for BiFeO_3_, PFeO_3_, and SbFeO_3_, respectively, for the majority spin channel. However, the maximum reflectivity in the minority spin channels was approximately 48%, 40%, and 46% for BiFeO_3_, PFeO_3_, and SbFeO_3_, respectively.

The following equation can be used to calculate the energy loss function L(ω) by utilizing the information of the dielectric function ε(ω) [58]:(12)$$L\left(\omega \right)=ln\left(-\frac{1}{\varepsilon (\omega )}\right)$$

The loss of rapidly moving electrons through the compounds can be determined by the energy loss function L(ω). Valuable information about the interaction of electronic systems with incident photons can be obtained from the spectra of L(ω). The calculated spectra of L(ω) for QFeO_3_ (Q = Bi, P, Sb) are presented in Figure 9 for both the minority and majority spins. The heights and shapes of the peaks in the spectra of L(ω) for QFeO_3_ (Q = Bi, P, Sb) are different in the minority and majority spin channels. The point form where peaks originate is known as the threshold energy of the material. The threshold energies of L(ω) calculated from Figure 9 for QFeO_3_ (Q = Bi, P, Sb) are presented in Table 3. The energy loss functions L(ω) and $\varepsilon_{2}(\omega )$ are inversely associated with each other, which means that L(ω) is active where $\varepsilon_{2}(\omega )$ is silent and vice versa.

The continuation of electrical transportation at higher energies of photons is known as optical conductivity. Optical conductivity σ(ω) is a very useful tool for the study of conducting materials. The following equation can be used to calculate σ(ω):(13)$$\sigma \left(\omega \right)=\sigma_{1}(\omega )+i\sigma_{2}(\omega )$$

Here, $\sigma_{1}(\omega )$ and $\sigma_{2}(\omega )$ are known as real and imaginary parts of optical conductivity. The following equation can be used to calculate optical conductivity σ(ω) by utilizing the information of the dielectric function ε(ω) [58]:(14)$$\sigma (\omega )=\frac{\omega }{4\pi }\varepsilon_{2}(\omega )$$

Transitions between CB and VB are used to calculate the optical conductivity of the aforesaid compounds. The calculated spectra of σ(ω) for QFeO_3_ (Q = Bi, P, Sb) are presented in Figure 10 for both the minority and majority spins. The heights and shapes of peaks in the spectra of σ(ω) for QFeO_3_ (Q = Bi, P, Sb) are different in the minority and majority spin channels. The point form where peaks originate is known as the threshold energy of the material. The threshold energies of σ(ω) calculated from Figure 10 for QFeO_3_ (Q = Bi, P, Sb) are presented in Table 3.

The following equation can be used to calculate I(ω) by utilizing the knowledge of $\varepsilon_{2}(\omega )$ and $\varepsilon_{1}\left(\omega \right)$ [58]:(15)$$I(\omega )=\sqrt{2\omega }{\left\{-\varepsilon_{1}(\omega )+\sqrt{\varepsilon_{1}^{2}(\omega )+\varepsilon_{2}^{2}(\omega )}\right\}}^{\frac{1}{2}}$$

The absorption coefficient $I\left(\omega \right)$ can be used to obtain information regarding the penetration length of the photon in the material or the length traveled by the photon inside the material before its complete absorption while having an energy greater than the energy band gap. It also tells us about the absorptive behavior of the compound. The calculated spectra of I(ω) for QFeO_3_ (Q = Bi, P, Sb) are presented in Figure 10 for both the minority and majority spins. The heights and shapes of peaks in the spectra of I(ω) for QFeO_3_ (Q = Bi, P, Sb) were different in the minority and majority spin channels. The point form where the peaks originate is known as the threshold energy of the material. The threshold energies of I(ω) calculated from Figure 10 for QFeO_3_ (Q = Bi, P, Sb) are presented in Table 3. The spectra of I(ω) showed increasing behavior with increasing energies of photons.

## 4. Conclusions

Scientists seek to fabricate novel multiferroic (MF) materials that exhibit interconnected properties such as piezoelectricity, magnetoelectricity, and magnetostriction and can operate effectively at room temperature. The effect of substituting pnictogen elements on optoelectronic and magnetic properties in BiFeO_3_ was investigated using first-principles-based DFT calculations. We noted that electronic states moved toward the Fermi level as we substituted Bi with P and Sb. The electronic band structures revealed that PFeO_3_/SbFeO_3_ are metallic compounds, whereas BiFeO_3_ is a semiconductor. The studied compounds are promising for spintronics, as they exhibit excellent magnetic properties. The calculated magnetic moments decreased as we replaced Bi with SB and P in BiFeO_3_. A red shift in the values of $\varepsilon_{2}(\omega )$ was evident from the presented spectra as we substituted Bi with Sb and P in BiFeO_3_. QFeO_3_ (Q = Bi, P, SB) showed the maximum absorption of incident photons in the visible region. The results obtained from calculating optical parameters suggest that these materials have a strong potential to be used in photovoltaic applications.

## Figures and Tables

**Figure 1 micromachines-14-02251-f001:**
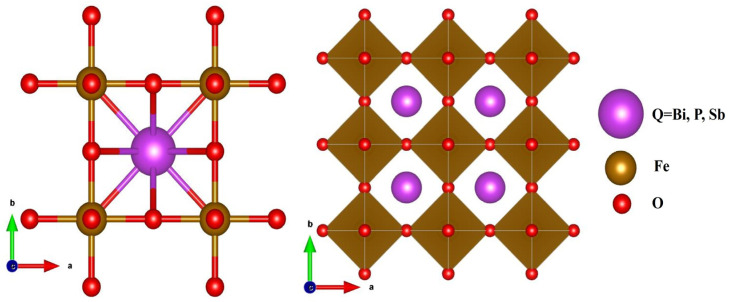
The crystalline structure of (a) BiFeO_3_, (b) PFeO_3_, and (c) SbFeO_3_.

**Figure 2 micromachines-14-02251-f002:**
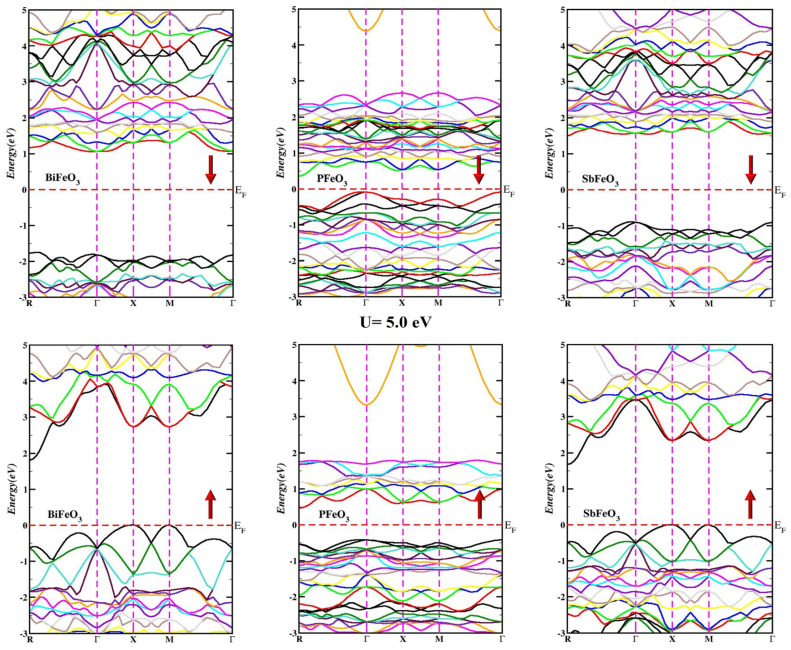
Energy band structure of QFeO_3_ (Q = Bi, P, Sb).

**Figure 3 micromachines-14-02251-f003:**
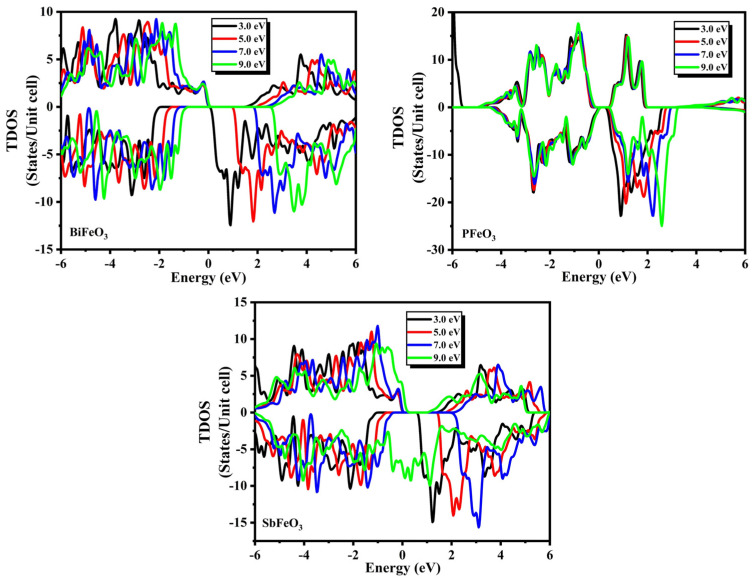
The total density of states for QFeO_3_ (Q = Bi, P, Sb) at U = 3.0, 5.0, 7.0, and 9.0 eV.

**Figure 4 micromachines-14-02251-f004:**
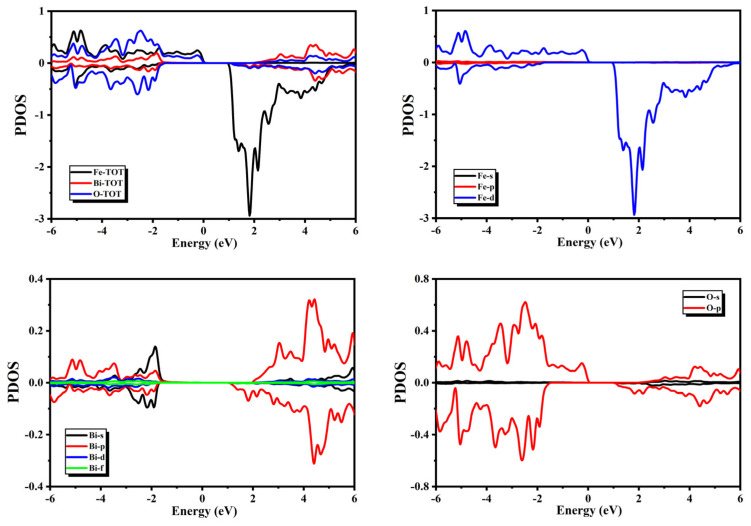
Partial density of states for BiFeO_3_.

**Figure 5 micromachines-14-02251-f005:**
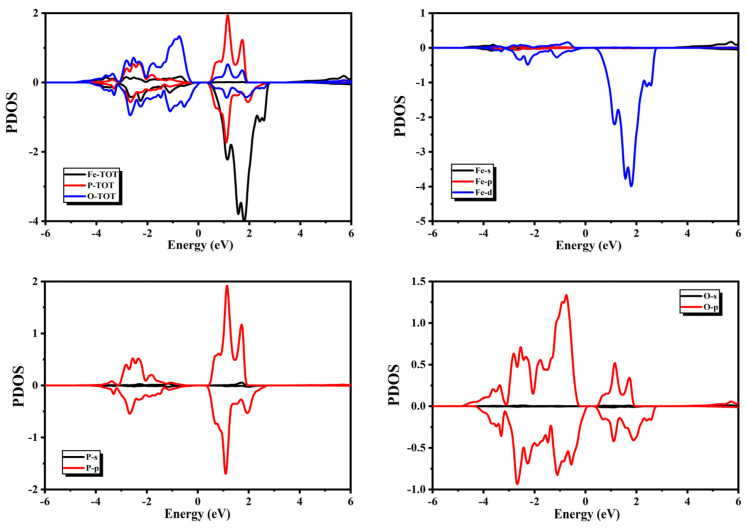
Partial density of states for PFeO_3_.

**Figure 6 micromachines-14-02251-f006:**
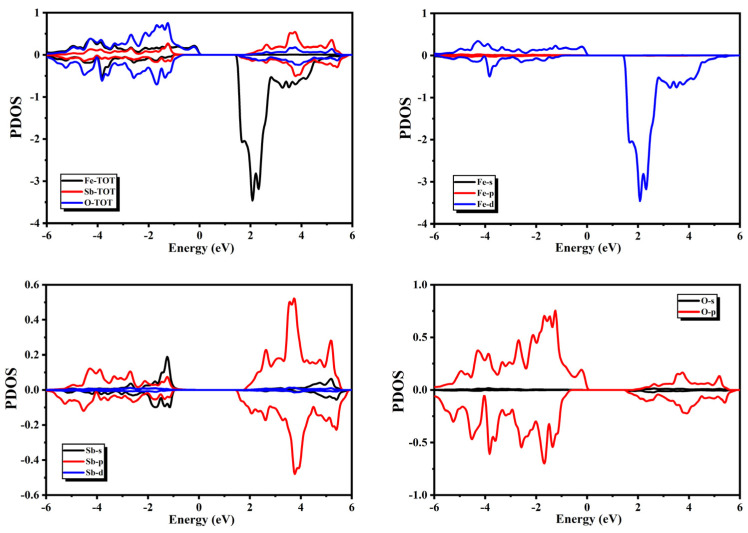
Partial density of states for SbFeO_3_.

**Figure 7 micromachines-14-02251-f007:**
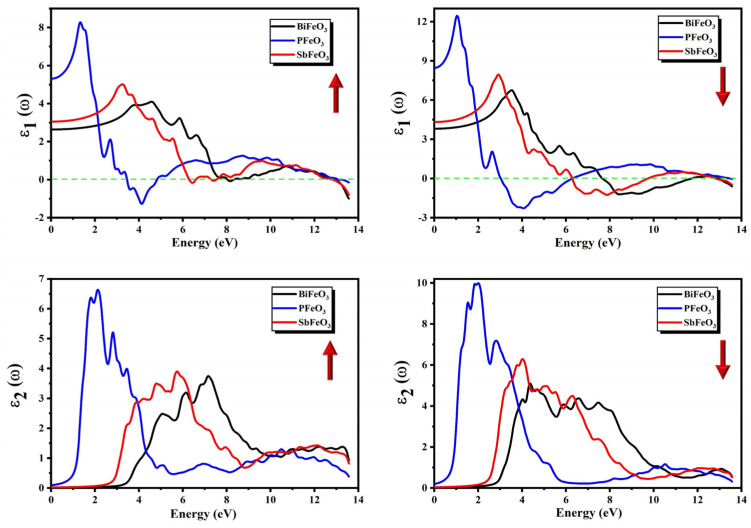
Calculated imaginary part $\varepsilon_{2}(\omega )$ and real part $\varepsilon_{1}(\omega )$ of complex dielectric function for QFeO_3_ (Q = Bi, P, Sb).

**Figure 8 micromachines-14-02251-f008:**
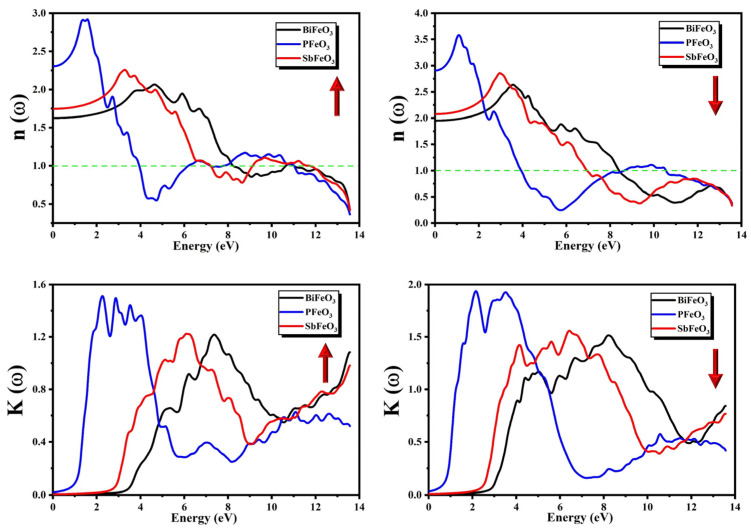
Calculated extinction coefficient K(ω) and refractive index n(ω) for QFeO_3_ (Q = Bi, P, Sb).

**Figure 9 micromachines-14-02251-f009:**
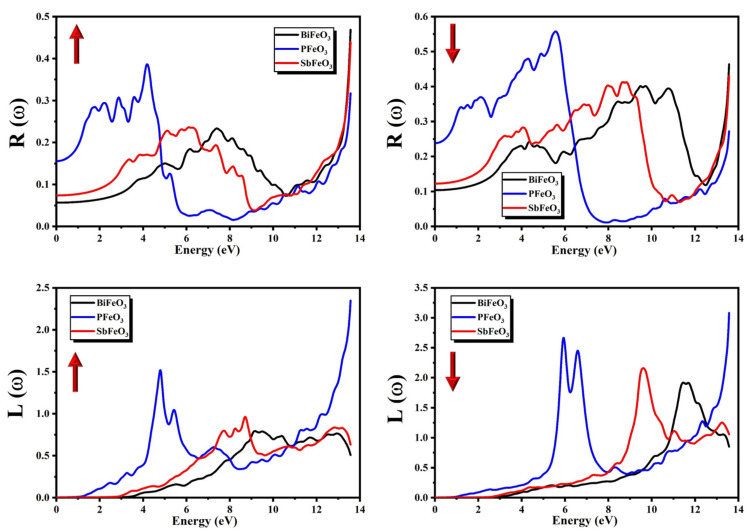
Calculated reflectivity coefficient R(ω) and energy loss function L(ω) for QFeO_3_ (Q = Bi, P, Sb).

**Figure 10 micromachines-14-02251-f010:**
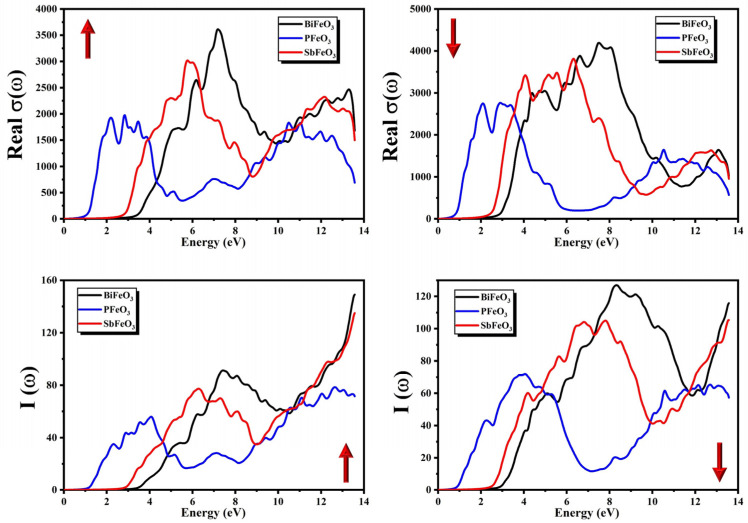
Calculated absorption coefficient I(ω) and real optical conductivity σ(ω) for QFeO_3_ (Q = Bi, P, Sb).

**Table 1 micromachines-14-02251-t001:** The calculated energy bandgaps (eV) for QFeO_3_ (Q = Bi, P, Sb) at various values of U.

Compounds	3.0 eV		5.0 eV		7.0 eV		9.0 eV	
	↑	↓	↑	↓	↑	↓	↑	↓
BiFeO_3_	1.4	2.2	1.87	2.62	2.2	3.3	2.2	3.48
PFeO_3_	0.82	0.30	0.87	0.39	0.46	0.24	0.47	0.82
SbFeO_3_	1.36	1.79	1.67	2.26	2.0	2.5	1.7	0

**Table 2 micromachines-14-02251-t002:** Partial and total magnetic moments of QFeO_3_ (Q = Bi, P, Sb).

Compound	$\mathrm{Magnetic}\;\mathrm{Moment}\;(\mu_{B})$				
BiFeO_3_	$m_{int}$	$m_{Bi}$	$m_{Fe}$	$m_{O}$	$m_{tot}$
	1.01300	0.03264	12.29616	1.65827	15.00008
PFeO_3_	$m_{int}$	$m_{P}$	$m_{Fe}$	$m_{O}$	$m_{tot}$
	0.40999	−0.10734	12.86157	1.83657	15.00081
SbFeO_3_	$m_{int}$	$m_{Sb}$	$m_{Fe}$	$m_{O}$	$m_{tot}$
	0.82277	0.03006	12.55617	1.59099	14.99998

**Table 3 micromachines-14-02251-t003:** Threshold energies (eV) of $\varepsilon_{2}(\omega )$, K(ω), I(ω), L(ω), and σ(ω) for QFeO_3_ (Q = Bi, P, Sb).

Compound	$\varepsilon_{2}\left(\omega \right)$		$K\left(\omega \right)$		I(ω)		$L\left(\omega \right)$		σ(ω)	
	↑	↓	↑	↓	↑	↓	↑	↓	↑	↓
BiFeO_3_	3.11	2.32	2.89	2.29	3.83	3.27	3.46	2.79	3.18	2.54
PFeO_3_	0	0	0.01	0.07	1.12	0.98	1.05	0.78	0.77	0.48
SbFeO_3_	2.42	1.94	2.21	1.92	2.97	2.85	2.77	2.44	2.27	2.01

**Table 4 micromachines-14-02251-t004:** Static values of $\varepsilon_{1}(0)$, n(0), and R(0) r QFeO_3_ (Q = Bi, P, Sb).

Compound	$\varepsilon_{1}\left(0\right)$		$n\left(0\right)$		R(0)	
	↑	↓	↑	↓	↑	↓
BiFeO_3_	2.64	3.81	1.62	1.95	0.057	0.104
PFeO_3_	5.31	8.46	2.30	2.91	0.156	0.238
SbFeO_3_	3.05	4.32	1.75	2.08	0.074	0.123

## Data Availability

Data is contained within the article.

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
