# Peer review of "Theoretical Justification of Structural, Magnetoelectronic and Optical Properties in QFeO3 (Q = Bi, P, Sb): A First-Principles Study"

_micromachines, 2023, doi:10.3390/mi14122251_

Round 1
Reviewer 1 Report (Previous Reviewer 1)
Comments and Suggestions for Authors
In this revised edition of the paper, the author includes more details in the methodology section, but I believe they do not support the majority of their choices. Once again, I insist on the U adjustment. Why 7.0 eV? The problem of DFT to describe correlated systems can be attributed to the tendency of xc functionals to over-delocalize valence electrons and to over-stabilize metallic ground states. That is why DFT fails significantly in predicting the properties of systems whose ground state is characterized by a more pronounced localization of electrons. Since the problem is rooted down to the band model of the systems, alternative models have been formulated to describe the correlated systems. One of the simplest models is the “Hubbard” model. One can intuitively conclude LDA+U is particularly dependent on the numerical value of the effective potential U. In may cases experimental bandgap is achieved with a U value near of 10 eV, whereas the crystal and electronic structures were better described with U < 5 eV. A value of 7 eV, in my opinion, is far too high for the author's intentions. In this respect, Why did the author choose this particular U value? Do they perform any sort of optimization? Can the author comment on the impact of their methodology on the outcomes in the conclusion? The paper can be improved if the author backs up t their choices.
Author Response
File Attached

Reviewer 2 Report (Previous Reviewer 2)
Comments and Suggestions for Authors
This revised manuscript reports a first principles study of electronic, magnetic and optical properties in QFeO3 (Q= Bi, P, Sb). Band structure, densities of states, magnetic and optical properties for BiFeO3 and SbFeO3 have been reported in literature in a large number of articles.
First-principles study of stability and polarization of SbFeO3 was reported by Guang Biao ZHANG et al. in Journal of the Physical Society of Japan 81 (2012) 074702, https://doi.org/10.1143/JPSJ.81.074702. This reference is not included to the revised manuscript.
The manuscript is full of flaws:
1. The authors clarified "The calculated values of energy bandgaps for BiFeO3, PFeO3 and SbFeO3 are 2.1, 0.46 and 2.2 eV, respectively." These values are strongly below the experimental values, see experimental articles.
2. The authors clarified "Rhombohedral crystalline structures for QFeO3 (Q= Bi, P, Sb) with space group R3c are presented in Figure 1.” This phase is the most investigated in the published articles, the corresponding results have been published for theoretical methods, optical calculations, experimental measurements. See Ceramics International, 45, 19793 (2019), Chemical Physics Letters 572, 78 (2013).
3. Partial density of states for Fe -d in Figures 4-6 are incorrect, because these states are ABOVE E_F, i.e., empty, also not corresponding to AFM, cf. figure 4 in the above paper by Guang Biao ZHANG et al. AFM ordering is ignored.
4. Code title WEIN2K" is corrected, the references 52 and 53 are not given as should be:
[52] P. Blaha, K. Schwarz, G. K. H. Madsen, D. Kvasnicka, J. Luitz, R. Laskowski, F. Tran and L. D. Marks, WIEN2k, An Augmented Plane Wave + Local Orbitals Program for Calculating Crystal Properties (Karlheinz Schwarz, Vienna University of Technology, Austria), 2018. ISBN 3-9501031-1-2
[53] P. Blaha, K. Schwarz, F. Tran, R. Laskowski, G. K. H. Madsen, and L. D. Marks, WIEN2k: An APW+lo program for calculating the properties of solids, J. Chem. Phys., 152, 074101 (2020).
The same computational codes are used as in the published papers. However, the calculations are done with mistakes. G-AFM ordering is ignored. The ferromagnetic ordering is considered instead. Metallic properties and electronic structure are obtained for the well-known insulators with the band gap of 2 eV. Magnetic moment of Fe is 13 mB which is known to be maximum 4.89 mB. Because of the wrong Fe moment, the total magnetic moments is also wrong. It is reported as 15.0-15.3 mB for QFeO3. Previously, it was calculated as 4 mB, see the references above.
There are mistakes in figures. In figures 4a-5a-6a, electronic Fe states are empty that is wrong. In table 2, magnetic moment of Fe is 13 mB which is known to be up to 4.89 mB. Because of the wrong Fe moment, the total magnetic moment is also wrong. It is reported as 15.0-15.3 mB for QFeO3. Previously, it was calculated as 4 mB, see the references above.
Comments on the Quality of English Language:
English should be improved.
Summing up the above points, this manuscript is not appropriate for publication. It should be rejected.
Comments on the Quality of English LanguageThis manuscript is not appropriate for publication. It repeats already published data with the incorrect results. It should be rejected again.
Author Response
File Attached.

Reviewer 3 Report (New Reviewer)
Comments and Suggestions for Authors
Parveen et al. presented a DFT study on the structural, electronic, magnetic, and optical characteristics of QFeO3 (Q= Bi, P, SB). They found that QFeO3 (Q= Bi, P, SB) shows maximum absorption of incident photons in 25 the visible region, which makes them great candidates in photovoltaic applications. The scientific method is sound, and the result is reliable. It is good to see a paper putting DFT calculations of all these three compounds together, which is interesting to the community. I therefore recommend publication in MDPI.
Author Response
File Attached

Round 2
Reviewer 1 Report (Previous Reviewer 1)
Comments and Suggestions for Authors
This new version addresses all of the previous issues. Accept in its current form
Reviewer 2 Report (Previous Reviewer 2)
Comments and Suggestions for Authors
This revised manuscript reports a first principles study of electronic, magnetic and optical properties in QFeO3 (Q= Bi, P, Sb). Band structure, densities of states, magnetic and optical properties for BiFeO3 and SbFeO3 have been reported in literature in a large number of articles.
First-principles study of stability and polarization of SbFeO3 was reported by Guang Biao ZHANG et al. in Journal of the Physical Society of Japan 81 (2012) 074702, https://doi.org/10.1143/JPSJ.81.074702. This reference is not included to the revised manuscript.
The manuscript is full of flaws:
1. The new values of energy bandgaps for BiFeO3, PFeO3 and SbFeO3 are about 1 or 1.5 eV from table 1. These values are strongly below the experimental values, see experimental articles.
2. The rhombohedral crystalline structures for QFeO3 (Q= Bi, P, Sb) is the most investigated in the published articles, the corresponding results have been published for theoretical methods, optical calculations, experimental measurements. See Ceramics International, 45, 19793 (2019), Chemical Physics Letters 572, 78 (2013). The references are ignored.
3. Partial density of states for Fe -d in Figures 4-6 are incorrect, because these states are ABOVE E_F, i.e., empty, also not corresponding to AFM, cf. figure 4 in the above paper by Guang Biao ZHANG et al. AFM ordering is ignored. Not changed.
4. Code title WEIN2K" is corrected, the references 52 and 53 are not given as should be:
[52] P. Blaha, K. Schwarz, G. K. H. Madsen, D. Kvasnicka, J. Luitz, R. Laskowski, F. Tran and L. D. Marks, WIEN2k, An Augmented Plane Wave + Local Orbitals Program for Calculating Crystal Properties (Karlheinz Schwarz, Vienna University of Technology, Austria), 2018. ISBN 3-9501031-1-2
[53] P. Blaha, K. Schwarz, F. Tran, R. Laskowski, G. K. H. Madsen, and L. D. Marks, WIEN2k: An APW+lo program for calculating the properties of solids, J. Chem. Phys., 152, 074101 (2020).
The same computational codes are used as in the published papers. However, the calculations are done with mistakes. G-AFM ordering is ignored. The ferromagnetic ordering is considered instead. Metallic properties and electronic structure are obtained for the well-known insulators with the band gap of 2 eV. Magnetic moment of Fe is 13 mB which is known to be maximum 4.89 mB. Because of the wrong Fe moment, the total magnetic moments is also wrong. It is reported as 15.0-15.3 mB for QFeO3. Previously, it was calculated as 4 mB, see the references above.
There are mistakes in figures. In figures 4a-5a-6a, electronic Fe states are empty that is wrong. In table 2, magnetic moment of Fe is 13 mB which is known to be up to 4.89 mB. Because of the wrong Fe moment, the total magnetic moment is also wrong. It is reported as 15.0-15.3 mB for QFeO3. Previously, it was calculated as 4 mB, see the references above.
Comments on the Quality of English Language:
English should be improved.
Again, this manuscript is not appropriate for publication. It should be rejected.
This manuscript is a resubmission of an earlier submission. The following is a list of the peer review reports and author responses from that submission.
Round 1
Reviewer 1 Report
Comments and Suggestions for Authors
This work provides a thorough examination of the electrical and optical properties of QFeO3 materials. The authors use the GGA+U DFT corrected method. The geometry and electrical properties at the GGA+U level are practically linearly dependent on the effective U value, which the authors chose and how they ensure the credibility of their results. On this particular point, a more detailed explanation is required in the methodological section. Saying that, the results appear interesting and have possible applications, and they deserve to be reported.
Main questions addressed: The computational evaluation of the structural, electrical, optical, and magnetic properties of QFeO3 (Q=Bi,P,Sb) perovkites materials in the search of nobel multiferroids material.
Relevance: The conclusion of the paper is that "these materials have strong potential to be used in photovoltaic applications." As a result, the electrical and optical qualities dictate the impact in the future. To put it another way, the relevance at this time is average. About the methodology: The author uses the GGA+U approach, however I'm not sure the U value they use or how they choose the U value. This is an important issue because the structural and electrical properties are determined by the U chosen, as are the optical properties. The authors must clarify how they chose the U parameter and whether or not this parameter was optimized; otherwise, the results are misleading. I believe that if the author explains this specific point in depth, the article will be considered for publication. References: Previous work in this topic was noted by the authors.
Reviewer 2 Report
Comments and Suggestions for Authors
In this first principles study the authors addressed electronic, magnetic and optical properties in QFeO3 (Q= Bi, P, Sb). Band structure, densities of states, magnetic and optical properties for BiFeO3 and SbFeO3 have been reported in literature in a large number of articles.
First-principles study of stability and polarization of SbFeO3 was reported by Guang Biao ZHANG et al. in Journal of the Physical Society of Japan 81 (2012) 074702, https://doi.org/10.1143/JPSJ.81.074702.
The manuscript is full of flaws:
1. GGA for semiconductors is incorrect. It results in a metal or underestimation of a gap. From the band structure in Figure 2, QFeO3 (Q= Bi, P, Sb) compounds are not insulators but metals. It is confirmed by Figure 3 Total density of states for QFeO3 (Q= Bi, P, Sb) with large DOS at the Fermi level.
The statement "in Figure 2, however, BiFeO3 is an indirect band gap semiconductor due to the presence of an energy band gap of 1.39 eV." is wrong because in Figure 2 and 3 BiFeO3 is a half-metal as can be found from the spin polarized band structure. It is not in agreement with experimental data and previous calculations suggesting BiFeO3 to be an AFM insulator with the band gap as 2 - 2.5 eV. Therefore, the results for Fermi surfaces are wrong.
2. Crystal structure type, space group and lattice parameters are not specified at all. One can assume: BiFeO3 (rhombohedral; R3c), CaTiO3 (perovskite prototype; cubic; Pm-3m), KNbO3 (tetragonal; P4mm), b-LaCoO3 (rhombohedral; R-3c), a-LaCoO3 (monoclinic; I2/a), or GdFeO3 (orthorhombic Pnma). The authors do not report. Structural properties are not investigated in this study.
3. Partial density of states for Fe -d in Figures 4-6 are incorrect, cf. figure 4 in the above paper by Guang Biao ZHANG et al. AFM ordering is ignored.
4. Code title WEIN2K" is incorrect, it should be WIEN2K. The authors cited incorrect references for the code. Correct references should be:
P. Blaha, K. Schwarz, G. K. H. Madsen, D. Kvasnicka, J. Luitz, R. Laskowski, F. Tran and L. D. Marks, WIEN2k, An Augmented Plane Wave + Local Orbitals Program for Calculating Crystal Properties (Karlheinz Schwarz, Vienna University of Technology, Austria), 2018. ISBN 3-9501031-1-2
P. Blaha, K. Schwarz, F. Tran, R. Laskowski, G. K. H. Madsen, and L. D. Marks, WIEN2k: An APW+lo program for calculating the properties of solids, J. Chem. Phys., 152, 074101 (2020).
5. All references in Section 2 are incorrect or self-citations. Inappropriate self-citations by authors are [52-58,60,62].
Summing up the above points, this manuscript is not appropriate for publication. It should be rejected.
Comments on the Quality of English LanguageModerate editing of English language required